# Hyperparametric solitons in nondegenerate optical parametric oscillators

Haizhong Weng[1,2], Xinru Ji [3], Mugahid Ali [1], Edward H. Krock[1], Lulin Wang[1], Vikash Kumar [1], Weihua Guo[4], Qing Wan[2], Tobias J. Kippenberg [3], John F. Donegan [1] ✉ & Dmitry V. Skryabin [5,6,7] ✉

Dissipative solitons and their frequency combs hold great potential for applications in optical communications, spectroscopy, precision time-keeping and beyond. Recent demonstrations based on the combination of second-harmonic generation and degenerate optical parametric oscillators (OPOs) show the interest in shifting soliton spectra away from the telecom's C-band pump sources. However, these approaches lack the tunability offered by nondegenerate OPOs. This work presents a proof-of-principle demonstration of solitons in a nondegenerate OPO system based on a silicon-nitride microresonator, with engineered dispersion and optimised coupling rates. By pumping a relatively low-Q resonance in the C-band, we excite a signal soliton comb centred around a far-detuned, high-Q, O-band resonance, as well as repetition-rate-locked combs at the pump and idler frequencies, with the latter occurring at a wavelength beyond 2 μm. The solitons supported by this platform − hyperparametric solitons − are distinct from other families of dissipative solitons, as they emerge when the narrow-band signal mode, phase-matched under negative pump detuning, reaches sufficient power to drive bistability in the parametric signal. We investigate the properties of hyperparametric solitons, including their parametrically generated background and multisoliton states, both experimentally and through theoretical modelling.

Microresonator optical frequency combs (microcombs) have made a profound impact on modern-day nonlinear integrated photonics and revolutionised fundamental research in optical solitons[1–4]. The unique features of such microresonator combs include their compactness and high repetition rates, with potential for mass production. At the same time, dissipative solitons represent the most application-relevant class of low-noise microcombs[2]. Broadening and stabilising soliton spectra have produced some of the most exciting research directions in this area. Spectra of solitons, typically centred around the pump wavelength, can span an octave in dispersion-engineered resonators[5–7] and may be further broadened by applying a second pump that also serves to stabilise the comb repetition rates and offset frequencies[8–10], and to initiate multi-colour soliton generation[11].

The rigid attachment of the aforementioned solitons, known as dissipative Kerr solitons[2], to a specific pump wavelength, which also builds the soliton background, may also limit the comb spectral bandwidth. Expanding the spectral range of dissipative solitons into the mid-infra-red region, as well as across other telecom bands, and into the visible and even UV parts of the spectrum, while still utilising

[1]School of Physics, CRANN, AMBER, and CONNECT, Trinity College Dublin, Dublin 2, Ireland. [2]Center for Heterogeneous Integration of Functional Materials and Devices, Yongjiang Laboratory, Ningbo, China. [3]Institute of Physics, Swiss Federal Institute of Technology Lausanne (EPFL), Lausanne, Switzerland. [4]Wuhan National Laboratory for Optoelectronics, and School of Optical and Electronic Information, Huazhong University of Science and Technology, Wuhan, China. [5]Department of Physics, University of Bath, Bath, UK. [6]Centre for Photonics, University of Bath, Bath, UK. [7]National Physical Laboratory, Teddington, UK. ✉e-mail: jdonegan@tcd.ie; d.v.skryabin@bath.ac.uk

the 1550 nm (C-band) pump, will enhance the impact of microcombs in telecommunications, spectroscopy and metrology.

By utilising nonlinear frequency conversion processes, e.g. Raman scattering, second and third harmonic generation, and parametric frequency conversion, it is possible to design microresonators that generate microcombs far away from the pump frequency. For example, Kerr solitons in silica glass microresonators that are pumped in the C-band have been shown to generate Stokes-pair solitons through the Raman effect, allowing tunability across the L and U communication bands of optical fibres[12,13]. Multi-colour solitons and combs with two pumps in Kerr resonators have also been demonstrated[8,9,14]. Second-harmonic generation in thin-film lithium niobate (LN) microresonators, which exhibit a combination of strong Kerr and second-order nonlinearities, has been employed to convert C-band solitons into non-solitonic near-visible combs[15] and to demonstrate two-colour solitons[16].

Parametric frequency conversion is, however, of primary interest in the context of our work. By engineering phase-matched parametric frequency conversion relying on either $\chi^{(2)}$ or Kerr nonlinearity, one can create optical parametric oscillators (OPOs) generating signals far detuned from the pump frequency[17-30], see Fig. 1a, b for a conceptual illustration of both degenerate and nondegenerate Kerr OPOs. In Kerr microresonator OPOs, the parametric gain is proportional to the coupled-in, i.e. intraresonator, pump power. As the gain increases, when the pump frequency is tuned closer to the resonance, the OPO signal can start coexisting with a stable non-OPO state, see Fig. 1b. In the case of degenerate OPOs, this coexistence has been a key prerequisite for demonstrations of parametric solitons making trains of signal pulses having a zero (non-OPO) background[31-33], see Fig. 1c, d.

In particular, parametric solitons have been demonstrated using $\chi^{(2)}$ nonlinearity for parametric down-conversion, from 775 nm pump to a 1550 nm signal, in an AlN microresonator[31]. A similar soliton experiment was later conducted in a fibre loop resonator, which included a short piece of $\chi^{(2)}$-active fibre for parametric gain[32]. The ref. 33 used an all-Kerr Si$_3$N$_4$ microresonator with the dispersion profile phase-matched for the degenerate four-wave mixing process, $2f_{signal} = f_{pump,1} + f_{pump,2}$, with the pumps close to 1.0 μm and 1.5 μm creating parametric gain for the degenerate or near-degenerate signal photons at 1.2 μm, see Fig. 1a–d. Here and below, $f$'s with various subscripts are used to annotate optical frequencies.

Common features of the parametric solitons observed in refs. 31–33 were device operation near the degeneracy point, providing a zero background for soliton pulses and the coexistence of solitons with identical intensity profiles and phases differing by $\pi$[32,33]. The zero-background property also means that the soliton central frequency does not dominate the rest of the signal comb, see Fig. 1d. This is unlike when a soliton is spectrally centred on the pump frequency and is governed by the classic form of the Lugiato-Lefever model[2].

Degenerate OPOs are efficient for frequency conversion of the pump light, but they lack the tunability of nondegenerate OPOs with significant spectral separation between signal and idler, which is achieved by controlling phase matching. Indeed, in degenerate OPOs or second-harmonic generation, adjusting the phase-matching condition only controls the conversion efficiency but not the frequency of the generated signal. In recent developments, nondegenerate OPOs operating in the continuous-wave (CW) regime within microresonators and waveguides have achieved improvements in both tunability and efficiency[17-30]. However, despite the wealth of results in the CW domain, no microresonator nondegenerate OPO that generates soliton microcombs has been demonstrated to date. Our current work fills this gap by presenting a new class of dissipative optical solitons and highlighting critical differences with solitons in degenerate OPOs.

In introducing the physical concept of our work, it is natural to start by mentioning that theoretical results on bright zero-background solitons in degenerate OPOs were reported in optics as early as two decades ago[34-37], and even earlier in the contexts of fluid mechanics and condensed matter physics[38-40].

Theoretical investigations of bright solitons in nondegenerate OPO systems—where the signal and idler fields are well separated spectrally—were also conducted during this early period[41,42]. These studies were technically and conceptually linked to the degenerate case and, therefore, found only solitons localised on a zero background, again arising in regimes where an OPO state coexists with a stable non-OPO state. Notably, we have not seen such solitons in the experiments reported below.

Here, we report a class of solitons in microresonator OPOs that have not been theoretically predicted or experimentally demonstrated. We use a Si$_3$N$_4$ resonator designed and pumped in C-band to support a nondegenerate phase-matched four-wave mixing process, $f_{signal} = 2f_{pump} - f_{idler}$, with the O-band signal near 1.25 μm and the idler

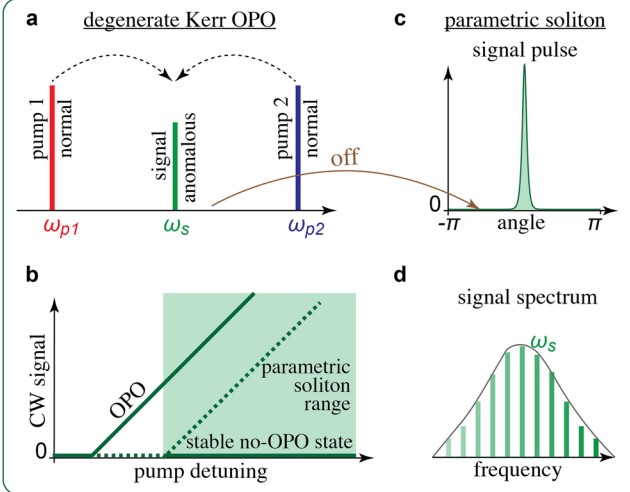

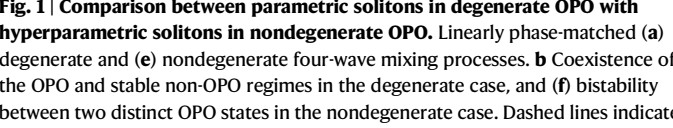

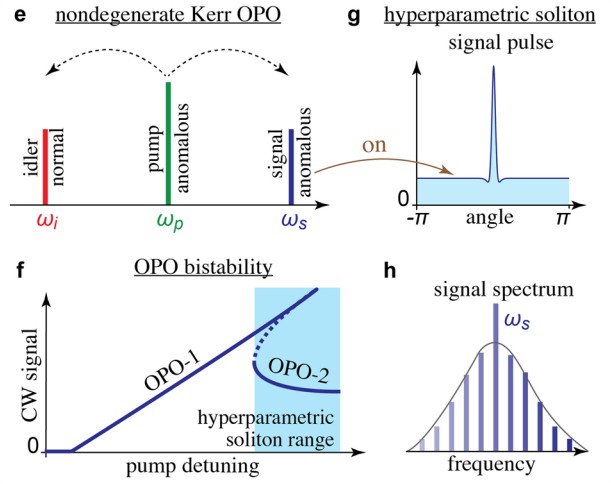

**Fig. 1 | Comparison between parametric solitons in degenerate OPO with hyperparametric solitons in nondegenerate OPO.** Linearly phase-matched (**a**) degenerate and (**e**) nondegenerate four-wave mixing processes. **b** Coexistence of the OPO and stable non-OPO regimes in the degenerate case, and (**f**) bistability between two distinct OPO states in the nondegenerate case. Dashed lines indicate unstable states. Parametric (**c**, **d**) and hyperparametric (**g**, **h**) solitons in the signal field and their respective spectra. Text in (**a**, **e**) also specifies whether the pump, $A_p$, signal, $A_s$ and idler, $A_i$, components fall into regions of normal ($D_2 < 0$) or anomalous ($D_2 > 0$) dispersion.

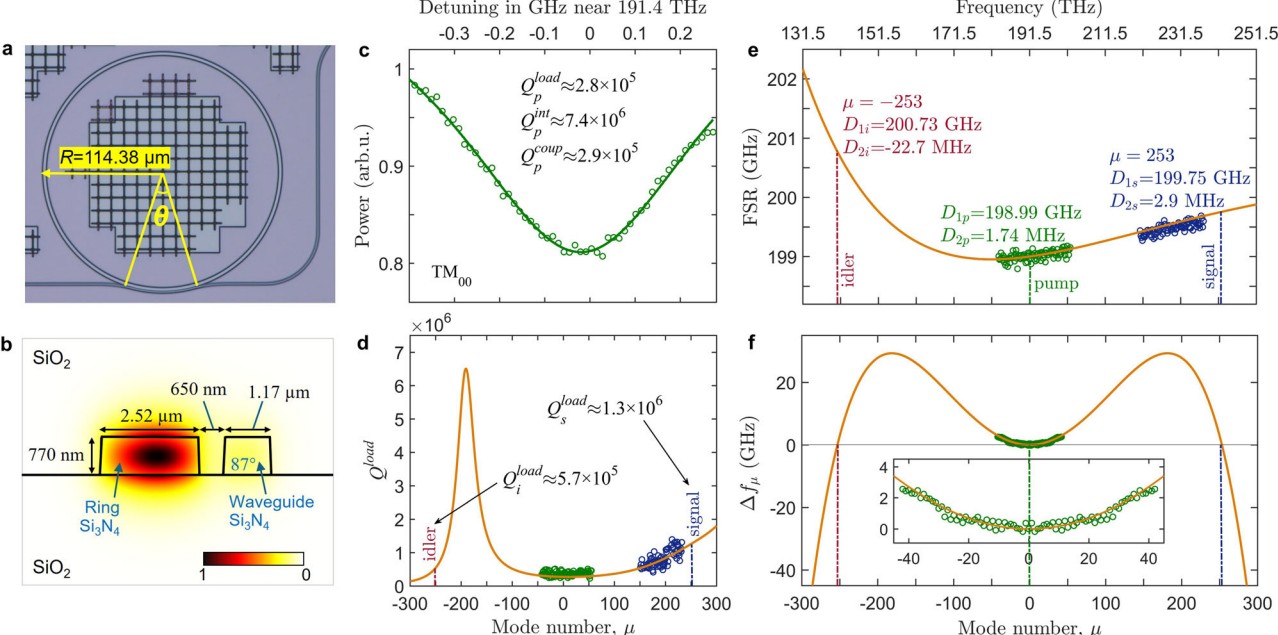

**Fig. 2 | Resonator geometry, loss, dispersion and phase matching profiles.**
**a** Microscope image of a ring resonator with a pulley coupler with $\theta = 25°$.
**b** Dimensions of the resonator and bus waveguide. **c** Measured resonance with
Lorentz fitting at the pump frequency of 191.4 THz, indicating a relative mode
number of zero ($\mu = 0$). $Q_p^{load}$, $Q_p^{intr}$ and $Q_p^{coup}$ are the loaded (total), intrinsic and
coupling quality factors, respectively. Close values of $Q_p^{load}$ and $Q_p^{coup}$ highlight
strong over-coupling conditions. **d** Simulated and experimental $Q^{load}$ versus rela-
tive mode number ($\mu$), showing a steep rise above $\mu = 150$ (220 THz) and a high
maximum corresponding to the anti-resonance coupling condition around
$\mu = -180$ (154 THz). **e**, **f** Free spectra range (FSR) and frequency mismatch ($\Delta f_\mu$, see
Eq. (1)) versus mode number and frequency. Dispersion is normal for decreasing
FSR and anomalous for increasing FSR. Circles in (**c**–**f**) show experimental points.

near 2 μm, see Fig. 1e. The resonator operates in the over-coupled
regime, where the pulley coupler is designed to ensure that the cou-
pling rate for the pump exceeds that for the signal and idler. We
demonstrate that such conditions ensure that the signal not only
increases with the pump but also depletes it sufficiently for the OPO
state to fold on itself, creating a bistable loop between two distinct
OPO states operating in the same resonator modes, see OPO-1 and
OPO-2 in Fig. 1f.

Then, we experimentally and numerically demonstrate that the
OPO bistability triggers the formation of signal solitons. This implies
that the parametric generation does not turn off at the soliton tails
since OPO-2 remains excited. This characteristic sets hyperparametric
solitons apart from the parametric ones observed in degenerate OPOs,
as illustrated in Fig. 1c, d, g, h. The term 'hyperparametric solitons',
which we coin here, emphasises that parametric processes play an
even more crucial role than in traditional 'parametric' solitons. The
latter are situated on a zero OPO background, while our solitons are
based on a fully developed monochromatic parametric signal. It is
worth noting that the term hyperparametric oscillations has been
previously used in the context of microresonators[19,43], but not in
relation to solitons.

The frequency of parametric solitons in the degenerate $\chi^{(2)}$ OPOs
is restricted to half of the pump frequency[31,32]. Similarly, the fre-
quency of Kerr parametric solitons hits the mid-point between the
two pumps[33]. Thus, both setups are tunable only by adjusting the
pump frequencies. In contrast, nondegenerate OPOs offer tuning
capabilities in the range of several tens to up to 100 THz by choosing
resonator geometries that provide suitable phase matching
conditions[20–24,28–30,44–47].

## Results

In this work, we used a Si₃N₄ microring resonator with a cross-section
of 770 nm × 2520 nm (thickness × ring width) and approximately
200 GHz repetition rate designed to provide phase-matching

conditions between two identical pump TM₀₀ photons with frequency
around 191.4 THz and signal and idler photons close to 242.0 THz and
140.8 THz, respectively. The details for designing the phase-matching
and choosing the present geometry condition can be found in
the Supplementary Information (Supplementary Fig. 1a, c). The reso-
nators were fabricated via a DUV-based subtractive approach, with the
capability of wafer-scale manufacturing of Si₃N₄ photonic integrated
circuits with ultra-low propagation loss and precise control of
dimensions[48], see 'Methods'. A pulley coupler with a gap of 650 nm was
used to provide over-coupling for the pump, see Fig. 2a, b. The
waveguide segment with the same curvature as the ring is approxi-
mately 50 μm long, which yielded the anti-resonance coupling range
around 154 THz corresponding to the drop in the coupling rate and a
sharp peak in loaded quality factor ($Q^{load}$)[49], see the simulated result
(solid curve) shown in Fig. 2d.

Available light sources allowed us to carry out the transmission
measurements in two regimes that cover C-band (pump) and O-band
(signal). Two transmission spectra are presented in Supplementary
Information (see Supplementary Fig. 1d), which confirm the over-
coupling of the TM₀₀ mode family in C-band with low extinction ratios
(<1 dB). With the resonances identified and fitted, the experimental
total quality factors (i.e. $Q^{load}$) for pump and signal are extracted and
depicted in Fig. 2d, showing good agreement with the simulation. $Q^{load}$
of the signal is $1.3 \times 10^6$, higher than that of the pump resonance
($Q_p^{load} = 2.8 \times 10^5$) near 191.4 THz, see Fig. 2c, d, which facilitates comb
generation to be triggered and dominated by the signal.

The resonant frequencies $f_\mu$ and the four-wave mixing phase
matching parameter,

$$\Delta f_\mu = f_\mu + f_{-\mu} - 2f_0, \quad \mu = 0, \pm 1, \pm 2, \ldots, \tag{1}$$

were computed numerically and plotted in Fig. 2e, f, coinciding with
the experimental data (circles) very well. The signal and idler fre-
quencies are determined from the condition of the parametric gain

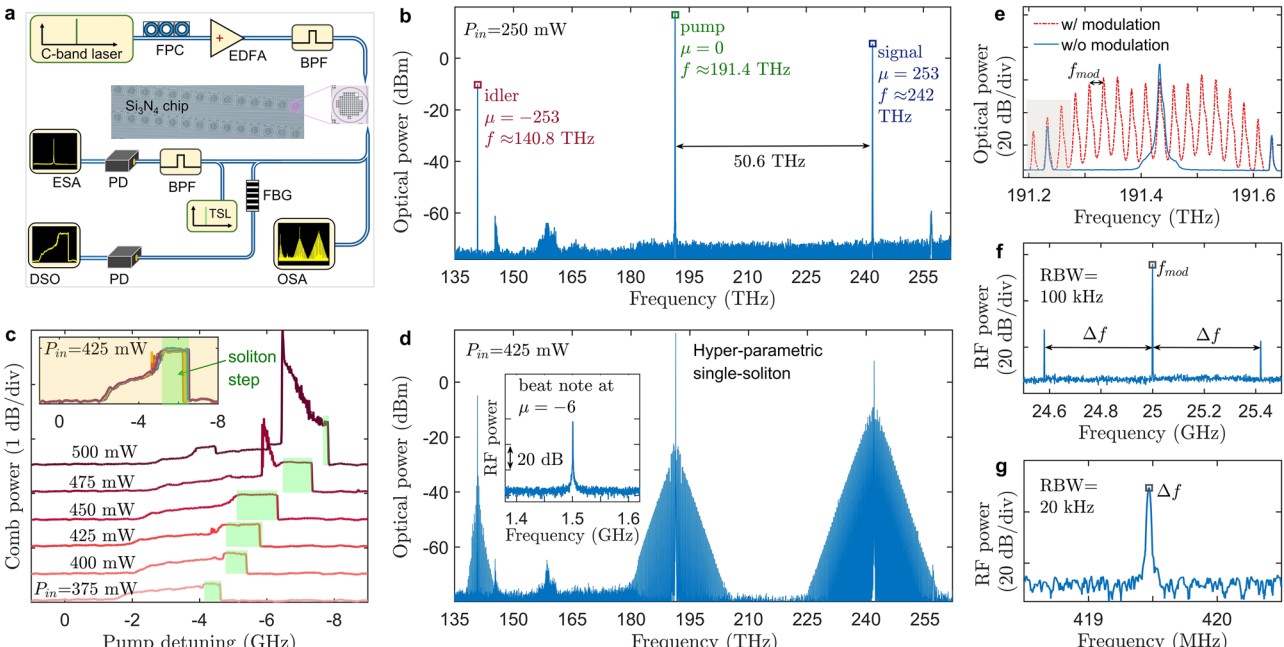

**Fig. 3 | Frequency comb measurement setup, soliton steps, OPO, soliton spectra and noise. a** Experimental setup. FPC polarisation controller, EDFA erbium-doped fibre amplifier, BPF band-pass filter, FBG fibre Bragg grating, PD photo-diode, ESA electrical spectrum analyser, TSL tunable semiconductor laser, DSO digital storage oscilloscope, OSA optical spectrum analyser. **b** Measured OPO spectrum under an on-chip power of 250 mW. **c** Measured OFC power when scanning the laser across the resonance at 191.4 THz for different values (375, 400, 425, 450, 475 and 500 mW) of the on-chip laser power. Green shading shows the soliton steps. The inset shows the ten repeat measurements in the 425 mW case. **d** Three-colour spectrum of a hyperparametric single-soliton. Inset: beat note between $\mu = -6$ soliton line and TSL. **e** The spectrum obtained by modulating the hyperparametric single-soliton with electro-optic (EO) modulators. **f, g** RF signals of the $\mu = -1$ comb line, see shaded interval in (**e**).

achieving its threshold value, which is most readily satisfied for a spread of modes in the proximity of the mode best complying with $\Delta f_\mu = 0$. Achieving $\Delta f_\mu = 0$ without invoking nonlinear frequency shifts is possible for large values of $\mu$, when there is a change between anomalous and normal dispersion. A plot of the phase-matching parameter vs mode numbers, $\mu$, for our resonator is shown in Fig. 2f, see also prior results in refs. 20–24,28–30. Defining dispersion as $D_{2\mu} = f_{\mu+1} + f_{\mu-1} - 2f_\mu$, we find that the groups of modes around the pump and signal have weakly anomalous dispersions, $D_{2p} = 1.7$ MHz and $D_{2s} = 2.9$ MHz, while the idler dispersion is large and normal, $D_{2i} = -22.7$ MHz. The FSRs of the pump, signal and idler are found to be $D_{1p} = 198.99$ GHz and $D_{1s} = 199.75$ GHz, and $D_{1i} = 200.73$ GHz, respectively.

The comb characterisation setup is schematically shown in Fig. 3a, where the amplified pump is coupled into the microresonator via a lensed fibre. For on-chip power of 250 mW, we observed the excitation of monochromatic signal and idler waves with the frequencies predicted by zeros of the phase-matching parameter, see Fig. 3b, broadly in agreement with prior work on Kerr OPOs[20–24,28–30]. We also tuned the signal and idler frequencies by tuning a range of various modes, see Supplementary Fig. 2. The output powers of the pump and signal in Fig. 3b are 17.4 dBm and 5.4 dBm, respectively, corresponding to an on-chip conversion efficiency of approximately 2%, which can be further improved by engineering the coupling rate and implementing temperature control. We note that since the signal is gaining energy from a product of the pump and idler amplitudes, with the latter being squared, the sufficiently high idler's Q is as critical as sufficiently high signal Q's for the generation of strong CW signals, see the Q versus mode number plot in Fig. 2d.

For on-chip pump powers ranging from 375 to 500 mW, we successfully observed hyperparametric soliton states corresponding to low-noise, three-colour frequency combs (see Figs. 3c–g, 4 and 5), which we describe in detail below. The microresonator chip was mounted on a temperature-controlled copper stage to enable tuning

of the phase-matching condition. Figure 3c presents comb power traces with the pump suppressed, as the pump wavelength was swept (0.5 nm/s) for six fixed on-chip pump power values. The green-shaded regions indicate soliton steps. A representative soliton spectrum (Fig. 3d) exhibits three triangular spectral bands centred at the idler, pump and signal frequencies, with the signal band showing the highest comb power by a significant margin.

A pronounced feature of the soliton spectra is the dominance of the signal comb power over the pump and idler combs. Specifically, the total power of the pump comb accounts for only 4% of the signal comb power, excluding the central lines of both combs. Coherence of the soliton state was confirmed via heterodyne detection between the comb line at $\mu = -6$ and a reference tunable semiconductor laser (TSL), yielding a narrow beat note, see Fig. 3d. To measure the soliton repetition rate, $f_{rep}$, we employed an electro-optic (EO) comb generator[50], see Supplementary Fig. 3. The comparison between spectra with and without modulation is shown in Fig. 3e. A spectral region near 191.2 THz was filtered and used for radio-frequency (RF) detection, see Fig. 3f, g. With a frequency offset $\Delta f$ of 419.5 MHz and a modulation frequency $f_{mod} = 25$ GHz, the repetition rate is $f_{rep} = 8 \times f_{mod} - \Delta f \approx 199.58$ GHz. This value closely matches the signal FSR in the linear regime, $D_{1s} = 199.75$ GHz, unambiguously indicating that the dominant signal pulse drives the repetition rate locking across the entire comb spectrum. By fitting the signal spectrum with a sech-squared function, we found that the soliton has a 3-dB bandwidth of 2 THz, giving a transform-limited pulse duration of 158 fs at full width half maximum.

The fact that the central signal mode at $\mu = 253$ dominates the comb spectrum generated around it should also be highlighted. This property contrasts sharply with that of parametric solitons in degenerate OPOs[31–33], which exhibit zero background and a central frequency that blends smoothly into the rest of the comb, cf. Fig. 1d, h. The strong central sideband in the signal comb in Fig. 3d should be compared with

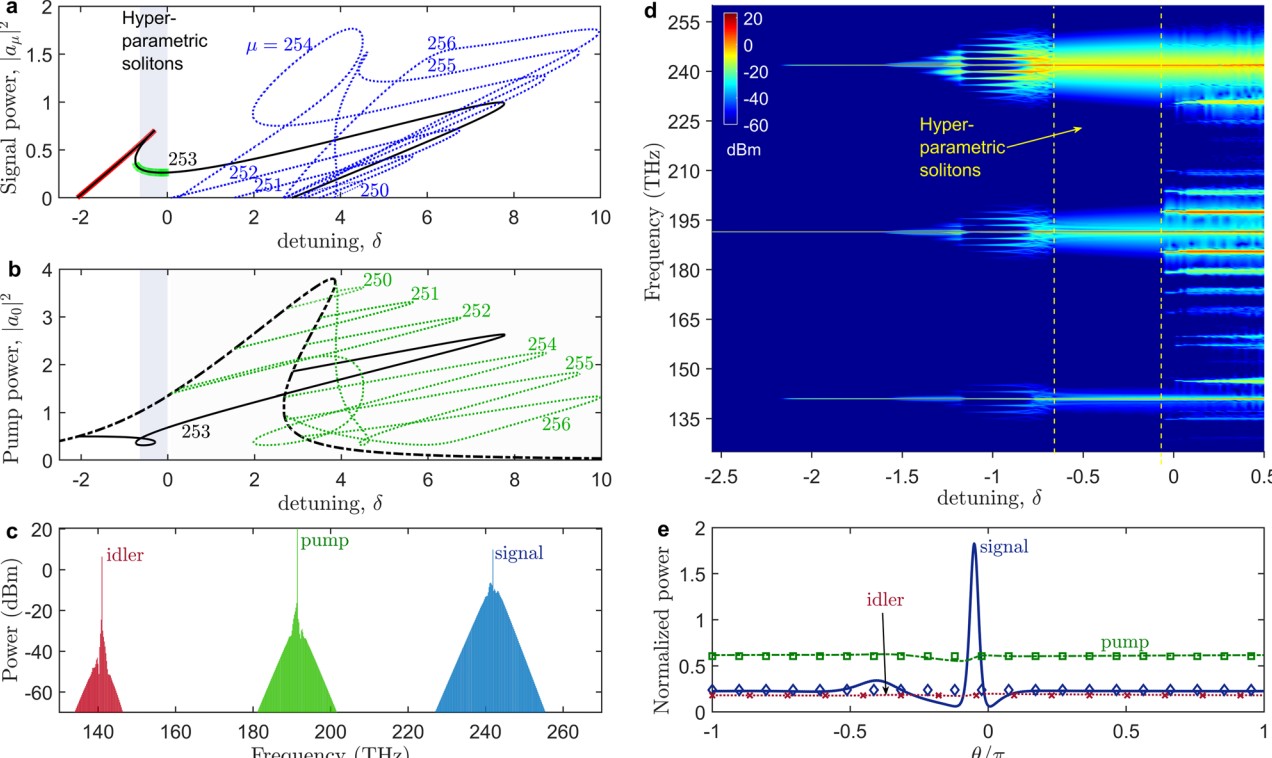

**Fig. 4 | Numerical results for parametric CW states and hyperparametric solitons. a** Intraresonator signal power, $|a_\mu|^2$, vs dimensionless detuning, $\delta = (f_0 - f_p)/\frac{1}{2}\kappa_0$, for six OPO states $\mu = \pm 250, \ldots, \pm 256$. $\mu = \pm 253$ OPO pair bifurcates first ($\delta \approx -2.15$) creating a bistable loop with itself; see the thick red and green lines highlighting the OPO-1 and OPO-2 states, cf. Fig. 1f. The shaded interval indicates the range of existence of hyperparametric solitons. **b** Dimensionless intraresonator pump power, $|a_0|^2$, corresponding to the signals shown in (**a**). The dashed-dotted line shows the nonlinear resonance without parametric generation, i.e. for $a_{\mu \neq 0} = 0$. **c** Three-colour output spectrum of a hyperparametric soliton computed for $\delta = -0.3215$. **d** Numerically computed spectra for a laser frequency sweep show parametric generation of the $\mu = \pm 253$ modes at 242 THz and 141 THz, which is then replaced by 3-colour combs and hyperparametric solitons. **e** Pulse profiles of the signal (bright soliton), pump (quasi-CW) and idler (quasi-CW) components of a hyperparametric soliton corresponding to the spectrum in (**c**). Pulse envelopes are computed as per Eq. (7) in 'Methods'. Squares, diamonds and crosses show the powers of the pump, signal and idler components for the OPO-2 state making up the soliton background. On-chip power applied in the modelling is $\mathcal{W} = 367$ mW. Scaling for the intraresonator power to use in **a**, **b**, **e** is 18.3 W, see 'Methods'.

no such feature in the parametric soliton data in Fig. 2a(iv) in ref. 31, Fig. 4d in ref. 32 and Fig. 4a, b in ref. 33. This effect is even more striking in the idler field at $\mu = -253$. Thus, we are dealing with multi-colour mode-locked pulses with a background shaped by the monochromatic parametrically generated signal and idler waves. This distinct behaviour motivates our use of the term hyperparametric soliton, as was already discussed in the previous section, see Fig. 1.

To test the tunability of hyperparametric solitons, we measured soliton generation across a range of resonators with varying geometries, each providing different phase-matching conditions. In particular, changing the ring width from 2530 to 2610 nm, while maintaining the pump at 191.1 THz, has led to a shift of the signal frequency by 2.3 THz, as shown in Supplementary Fig. 4a. We have further pumped a sequence of resonances in the interval from 191.4 to 193.6 THz and tuned the signal frequency from 242 to 249.7 THz, as shown in Fig. S4b. Correspondingly, the idler frequency was adjusted from 140.8 to 137.8 THz. The EDFA's range was the main factor limiting the observed tuning range.

Since no present concept or theory explains our measurements, it appears instructive to outline our numerical results and their interpretations. The first step in our analysis of mechanisms of the hyperparametric soliton generation was to solve the problem of three-wave parametric interaction of the $\mu = 0$ mode with several pairs of the signal and idler in the proximity of the $\Delta f_\mu = 0$ point, see Methods. Numerically computed signal components of all OPO states found above the parametric threshold for a chosen input power are shown in Fig. 4a. The pump, $\mu = 0$ mode, becomes strongly depleted by the generated

signal and idler pairs so that its dependence on the laser frequency deviates from the parametric-process-free nonlinear resonance shown by the dashed line in Fig. 4b.

Notably, generating $|\mu| = 253$, 254 and 255 states leads to the bistable OPOs. One should note that the negative-detuning range of the 253 state stays isolated from the range of detunings where multiple other parametric states coexist, see Fig. 4a and Fig. 1f. We name the upper and lower branches of the bistability loop made by this state as OPO-1 and OPO-2. However, the instabilities of OPO-1 and OPO-2 relative to the excitation of other modes (modulational instability) can not be excluded.

To check stability and instability scenarios, we modelled a system of coupled equations for the amplitudes $a_\mu$ of the resonator modes $\mu = -512, \ldots, 0, \ldots, 511$, covering the spectrum from 90 to 290 THz, see 'Methods'. We have found that the OPO-1 is typically modulationally unstable, whereas OPO-2 is typically stable over the detuning interval during which they coexist, a condition that leads to the generation of hyperparametric solitons. The spectral carpet shown in Fig. 4d represents a bifurcation diagram computed by varying the laser frequency from large negative detunings, $\delta = (f_0 - f_p)/\frac{1}{2}\kappa_0$, to positive ones, illustrating the modulational instability and soliton generation processes. Here, $f_p$ is the laser frequency and $\kappa_0$ is the width of the pumped resonance. CW parametric generation begins at $\delta \approx -2.16$ and destabilises due to sideband generation at $\delta \approx -1.6$, forming three-colour frequency combs. These then evolve into hyperparametric solitons that exist for $\delta$ values in the $(-0.65, -0.06)$ range.

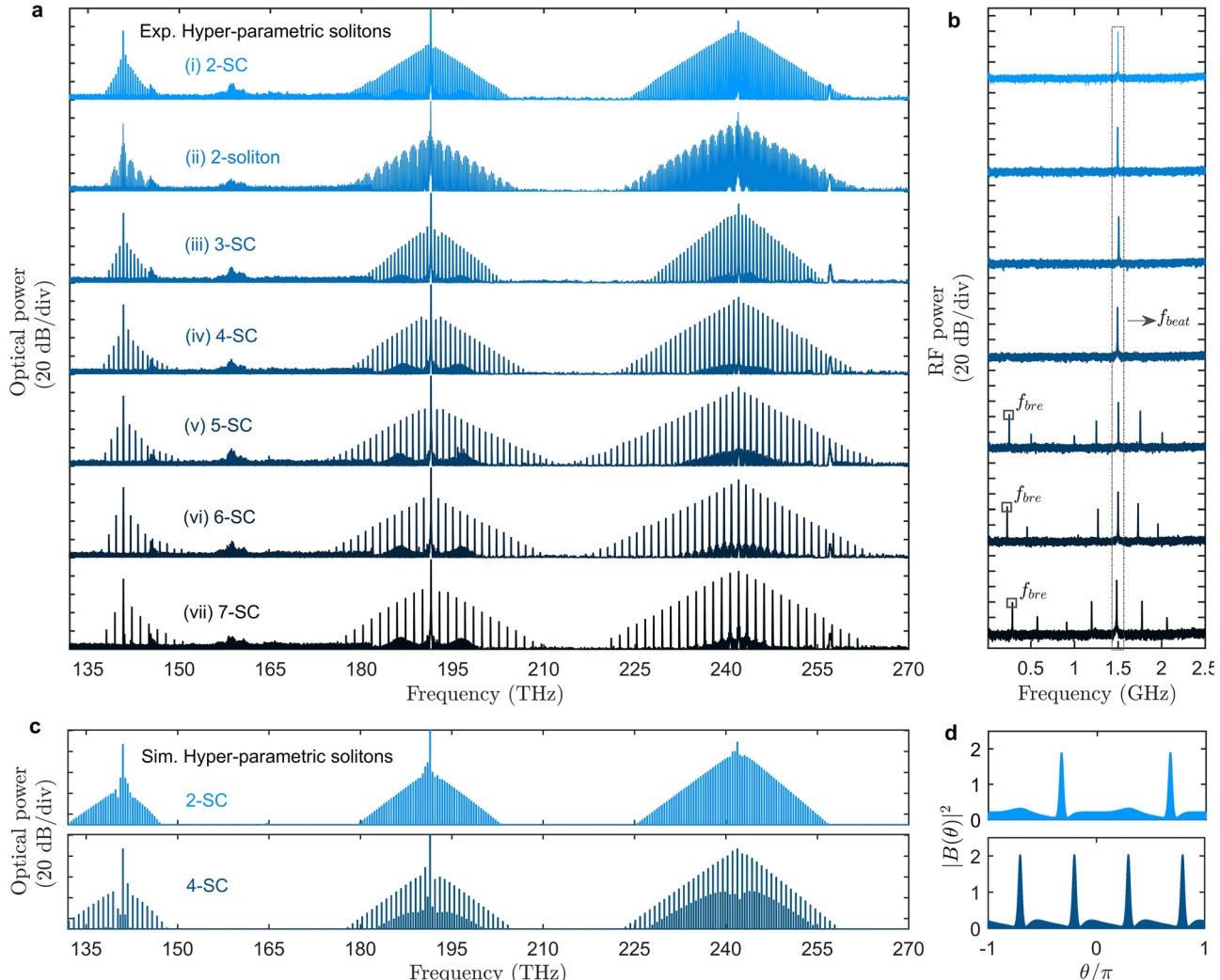

**Fig. 5 | Hyperparametric soliton crystals, quasi-crystals, multi-soliton states and breathers. a** Experimentally measured three-colour spectra (top to bottom) for the two-soliton crystal, two-soliton state, three-soliton crystal, four-soliton crystal, and five-, six- and seven-breather-soliton crystals. **b** Experimental hetero-dyne beat note measurements (between a comb line and TSL) for the soliton states shown in (**a**). Squares in the three bottom plots indicate the breather frequencies. **c** Numerically modelled spectra of a two-soliton crystal and a four-soliton quasi-crystal. **d** Simulated spatial profiles of signal pulses associated with the 242 THz spectra in (**c**).

A numerically computed spectrum of the hyperparametric soliton (see Fig. 4c) qualitatively matches the typical experimentally observed spectra in Fig. 3d. The reconstructed pulse envelope profiles for a hyperparametric soliton's signal, pump and idler components are shown in Fig. 4e. It shows that the pump and idler waves are quasi-CW fields, and only the signal is a pronounced bright soliton resting on a background with finite amplitude.

The idler is not expected to make a pulse because its frequency lies in the interval of large normal dispersion. While the pump is in the anomalous dispersion range, it experiences a loss rate three times higher than the signal and also remains quasi-CW. For the pump to trigger soliton generation of its own, i.e. without the parametric process involved, requires the laser frequency to be tuned closer to the tip of its resonance ($\delta \approx 4$ on the dashed black line in Fig. 4b).

When spectra of the pump, signal and idler combs remain well isolated from each other, one can split the full coupled-mode system into equations for three modal groups and introduce the respective envelope functions, $A_p$, $A_s$ and $A_i$, Fig. 4 caption and Methods. By checking the signal equation, see Eq. (9) in Methods, reveals that the signal soliton centred on a mode $\mu'$ is excited by the quasi-CW

nonlinear polarisation wave induced by the four-wave mixing process $A_p^2 A_i^* \approx a_0^2 a_{-\mu'}^*$.

Thus, the signal equation can be considered the generalised Lugiato-Lefever equation, where the pump has a complex dependence on the detuning introduced by the parametric process. The latter is anticipated to be the OPO-2 state. Diamonds, squares and crosses in Fig. 4e show perfect matching between the hyperparametric soliton tails and the power values of the signal, pump and idler of the OPO-2 state, see Fig. 4a. This finding unambiguously confirms that the hyperparametric soliton has the background wave matching the OPO-2 state.

By pumping different resonator modes, tuning pump power and varying the phase matching parameter, we have further checked that conditions for the hyperparametric solitons to exist are that the first parametric state bifurcating from the pump for the negative detunings must become bistable and provide a range of parameters (pump power and detuning) where the lower (OPO-2) branch is stable and the upper (OPO-1) branch is modulationally unstable. These conditions are typically met when the net signal losses are sufficiently small and the

power of the four-wave mixing term, $a_0^2 a_{-\mu'}^*$, driving the signal mode $a_{\mu'}$, is sufficiently high.

For the soliton generation, we recorded a video (see Supplementary Video 1) showing how the signal and pump spectra evolve during manual frequency tuning across the hyperparametric soliton range. Initially, non-solitonic combs replace the monochromatic signal and pump, which then evolve into soliton generation. The strong, ≈8 dBm, central sideband in the signal component rising above the comb starting around −8 dBm, unambiguously points to the hyperparametric regime. The soliton existence on the negatively detuned tail of the resonance, where intraresonator fields and thermal effects are modest, enables the simplicity of the manual tuning. Tuning to the range of positive detunings boosts the circulating power, leading to the sudden loss of resonance due to thermal shifts. Experimental access to these regimes may be the subject of future work using one of the established techniques for observing positively detuned non-parametric solitons of the Lugiato-Lefever equation[2,51].

We also observed the generation of spectrally broad signals and idler combs at relatively high pump powers, see Supplementary Fig. 5. For large detuning, a sparse Turing-pattern-like OPO comb has a line separation of 11 FSR, which then gradually transitions to combs with broader, densely filled spectra. The pump-only Turing-pattern-like comb with the 28 FSR line-to-line separation is seen at the end of this experimental sequence. This level of pump power already takes us outside the hyperparametric soliton range, while pure Kerr solitons can be found numerically at these powers for red detuning. They are, however, not observed experimentally due to the lack of measures to control thermal effects in our setup.

The measured spectra can vary from a single soliton to quasi-crystals or soliton crystals by slightly changing the pump power, as shown in Fig. 5a. In crystals containing $N$ solitons, only every $N$th mode is excited, while quasi-crystals exhibit weaker lines between the strong spectral peaks. With the same resonator, we successfully excited the soliton crystals with $N = 2, 3, 4, 5, 6, 7$. Additionally, multi-soliton non-crystal states, where $N$ pulses are situated far apart from the $2\pi/N$ separation distances, have also been frequently observed, see the two-soliton in state (ii). A single beat note near 1.5 GHz in heterodyne measurements shown in Fig. 5b confirms that the two-, three- and four-soliton states are completely coherent. However, the five-, six- and seven-soliton crystals are typically the breather states[52,53].

Experimentally, we have observed the OPO comb spectra with signal and idler tuned between 230 and 249 THz (1.3–1.2 μm) and 138–150 THz (2.2–2 μm), respectively. At the extremities of this range, a broadband OPO comb was generated but not a soliton state due to the unoptimized coupling rate. While dispersion engineering allows extremely broad-band phase matching[22], the existence of multi-colour solitons imposes its own constraints on the balance of dispersion and quality factors for which a complete understanding should be the subject of future work.

As the number of hyperparametric solitons in a crystal increases, the magnitude of the central sidebands in the idler and pump fields remains dominant. Conversely, the sidebands in the signal field decrease significantly. This is because the signal background rapidly loses power as the ring gets filled with pulses. Meanwhile, the pump and idler fields remain quasi-cw, with their power being only weakly dependent on the number of solitons in the crystal. This behaviour has been observed in the experimental and numerical data, see Fig. 5a–d.

Soliton crystals have now emerged as a sub-field that connects photonics, condensed matter and topological physics[54–59], while the multi-soliton and crystal studies included parametric systems[37,60], the hyperparametric frequency conversion effect paves a new path for their investigation.

## Discussion

We demonstrated hyperparametric three-colour idler-pump-signal solitons in a nondegenerate Kerr microresonator OPO with a 200 GHz pulse repetition rate. The idler and pump soliton components exhibit quasi-CW behaviour, while the signal component forms a pronounced bright soliton pulse sitting on a parametrically generated background. This soliton regime arises when one of the above-threshold signal-idler pairs makes a hysteresis loop (bistability) from the modulationally unstable high-power and stable low-power OPO states. Our modelling shows that this OPO bistability occurs at negative pump detunings, which avoids the positive detuning region typically associated with pump-centred Lugiato-Lefever solitons. At negative detunings, intra-resonator powers remain relatively low, minimising unwanted thermal effects. The quasi-CW nature of the pump and idler components is maintained through a combination of strong coupling losses at the pump (overcoupled resonator) and normal dispersion at the idler wavelength.

Nondegenerate OPOs provide broad and flexible tunability through resonator geometry and temperature control of the refractive index and phase-matching conditions[20–24,28–30]. This contrasts with degenerate OPOs, in which the signal frequency is strictly determined by the pump[31–33]. We also observed and reported the formation of crystals and breather states of hyperparametric solitons. Specifically, we generated solitons in the optical communication O-band using a C-band pump source. Expanding the availability of O-band light sources based on silicon photonics is a key milestone in the ongoing development of low-power optical interconnects for data centre applications[61].

Overall, our results on hyperparametric solitons open new avenues for studying their dynamics and interaction with other soliton types in nondegenerate Kerr and especially in $\chi^{(2)}$ microresonators, where realising comb generation around widely separated and tunable signal and idler frequencies remains an open challenge.

## Methods
### Device fabrication and characterisation
The Si$_3$N$_4$ microresonators were fabricated using a deep ultraviolet (DUV) based subtractive process on 4-inch silicon wafers. A 4 μm thermal oxide layer was first grown as the bottom cladding. To mitigate the high tensile stress in thick Si$_3$N$_4$ films, interconnected stress-release trenches (3.5 μm depth) were patterned in the SiO$_2$ substrate using dry etching, spaced 10–30 μm from waveguide regions to ensure waveguide uniformity. A 770 nm thick stoichiometric Si$_3$N$_4$ layer was then deposited in a single low-pressure chemical vapour deposition (LPCVD) step, achieving 0.6% thickness uniformity across the wafer. Waveguides were patterned using KrF 248 nm DUV stepper lithography with 180 nm resolution, followed by anisotropic dry etching with a C$_x$F$_y$ chemistry. An LPCVD SiO$_2$ hard mask was employed during etching to achieve smooth sidewalls with 87° sidewall angles. The devices underwent high-temperature annealing at 1200 °C for 11 h in N$_2$ atmosphere to reduce hydrogen-related absorption losses. A 1.3 μm thick LPCVD SiO$_2$ top cladding was then deposited and subjected to identical annealing conditions.

### OFC measurements
To characterise the resonator's transmission, we swept two separate tunable lasers across the 1480–1640 nm and 1260–1360 nm wavelength ranges with a high resolution of 0.1 pm and recorded the output optical power. These transmission spectra enabled the extraction of key resonator parameters, including resonance linewidths, free spectral ranges (FSRs) and phase-matching conditions. For hyperparametric soliton generation, we used a C-band tunable laser source, amplified by an erbium-doped fibre amplifier (EDFA), which was coupled into the waveguide through a lensed fibre. Before the lensed fibre, a fibre polarisation controller (FPC) was employed to optimise the

pump polarisation and a bandpass filter (BPF) was inserted to suppress amplified spontaneous emission (ASE) noise.

The output light was collected using a second lensed fibre and split into two detection paths. One path, carrying the majority of the signal, was directed to optical spectrum analysers (OSAs) for spectral monitoring. Simultaneously, the other path was routed either through a fibre Bragg grating (FBG) for comb power monitoring using a digital storage oscilloscope (DSO), or through a bandpass filter (BPF) and combined with an auxiliary TSL for radio-frequency (RF) noise analysis using an electrical spectrum analyser (ESA). Access to optical parametric oscillation (OPO) sidebands and hyperparametric soliton states was achieved by gradually tuning the pump laser into resonance, either through continuous wavelength sweeping or manual adjustment. The characteristic soliton dynamics are illustrated in the Supplementary Video 1, which captures the manual tuning of the pump wavelength from 1566.37 nm to 1566.42 nm at an on-chip pump power of 425 mW.

## Nonlinear modelling

Multimode intra-resonator field is expressed as

$$\mathcal{A}e^{iM\vartheta - i\omega_p \tilde{t}} + c.c., \quad \mathcal{A} = \sum_{\mu} a_{\mu}(\tilde{t})e^{i\mu\vartheta}. \tag{2}$$

Here, $\vartheta = (0, 2\pi]$ is the angular coordinate in the laboratory frame, $\tilde{t}$ is time and $a_{\mu}$ are mode amplitudes and $\mu = 0, \pm 1, \pm 2, \ldots$ is the relative mode number. Coupled-mode equations governing nonlinear interactions between $a_{\mu}$ are [62–65]

$$i\partial_t a_{\mu} = -\gamma \sum_{\mu_1 \mu_2 \mu_3} \hat{\delta}_{\mu, \mu_1 + \mu_2 - \mu_3} a_{\mu_1} a_{\mu_2} a_{\mu_3}^*$$
$$+ (f_{\mu} - f_p)a_{\mu} - \frac{\kappa_{\mu}}{2}\left(a_{\mu} - \hat{\delta}_{\mu, 0}\sqrt{b\mathcal{W}}\right). \tag{3}$$

Here, $\hat{\delta}_{\mu, \mu'} = 1$ for $\mu = \mu'$ and is zero otherwise. The resonator spectrum, $f_{\mu} = \omega_{\mu}/2\pi$, and the mode number dependent linewidth, $\kappa_{\mu}$, were computed using Lumerical software, see Fig. 2. Since we work with frequencies, $f_{\mu}$, and not the angular frequencies, $\omega_{\mu}$, time $t$ in Eq. (3) was redefined as $t = 2\pi\tilde{t}$. The pump laser frequency, $f_p$, is tuned around $f_0 = 191.44$ THz corresponding to the $M = 820$ resonance. $\mathcal{W}$ is the on-chip laser power, $b = 179.5$ is the resonator power build-up factor for $f_p = f_0$. $\gamma = f_0 n_2/Sn_0$ is the nonlinear parameter[15]; $n_2 = 2.6 \cdot 10^{-19}$ m$^2$/W (Kerr coefficient), $S = 1.32 \cdot 10^{-12}$ m$^2$ (mode area), $n_0 = 2.022$ (refractive index), $\gamma = 18.65$ MHz/W. $|a_{\mu}|^2$ represents intrraresonator powers of the comb lines scaled to have units of Watts[64]. Data shown in Figs. 3c–e, 4 are obtained by modelling Eq. (3). While numerically solving Eq. (3), we divided them by $\kappa_0/2$, $\kappa_0 = 683$ MHz. The scaling parameter applied to plot dimensionless modal powers in Fig. 4a, b, e is $\kappa_0/2\gamma = 18.3$ W. Linewidth parameters are $\kappa_{\mu} = f_{\mu}/Q_{\mu}^{\text{load}}$, see Fig. 2d and Supplementary Table I.

Three-wave OPO states, represented by the pump mode and a signal-idler pair, see Fig. 4a, b, were computed from the three-mode reduction of Eq. (3).

$$i\partial_t a_0 = -2\gamma a_{\mu} a_{-\mu} a_0^* - \gamma(|a_0|^2 + 2|a_{\mu}|^2 + 2|a_{-\mu}|^2)a_0$$
$$+ (f_0 - f_p)a_0 - i\frac{\kappa_0}{2}(a_0 - \sqrt{b\mathcal{W}}),$$
$$i\partial_t a_{\mu} = -\gamma a_0^2 a_{-\mu}^* - \gamma(|a_{\mu}|^2 + 2|a_0|^2 + 2|a_{-\mu}|^2)a_{\mu}$$
$$+ (f_{\mu} - f_p)a_{\mu} - i\frac{\kappa_{\mu}}{2}a_{\mu},$$
$$i\partial_t a_{-\mu} = -\gamma a_0^2 a_{\mu}^* - \gamma(|a_{-\mu}|^2 + 2|a_0|^2 + 2|a_{\mu}|^2)a_{-\mu}$$
$$+ (f_{-\mu} - f_p)a_{-\mu} - i\frac{\kappa_{-\mu}}{2}a_{-\mu}. \tag{4}$$

Parametric terms are placed first after the equal signs in all three equations. The role of phase-matching parameter, $\Delta f_{\mu}$, see Eq. (1), in the three-wave model becomes explicit on the observation that

$$f_{\pm\mu} - f_p = \frac{1}{2}\Delta f_{\mu} + (f_0 - f_p) \pm \zeta_{\mu}, \tag{5}$$

while the $\zeta_{\mu} = (f_{\mu} - f_{-\mu})/2$ term can be eliminated from Eq. (4) by substituting $a_{\pm\mu} = \tilde{a}_{\pm\mu}\exp(\mp i\zeta_{\mu}t)$. So that, Eq. (4) becomes

$$i\partial_t a_0 = -2\gamma\tilde{a}_{\mu}\tilde{a}_{-\mu}a_0^* - \gamma(|a_0|^2 + 2|\tilde{a}_{\mu}|^2 + 2|\tilde{a}_{-\mu}|^2)a_0$$
$$+ (f_0 - f_p)a_0 - i\frac{\kappa_0}{2}(a_0 - \sqrt{b\mathcal{W}}),$$
$$i\partial_t \tilde{a}_{\mu} = -\gamma a_0^2 \tilde{a}_{-\mu}^* - \gamma(|\tilde{a}_{\mu}|^2 + 2|a_0|^2 + 2|\tilde{a}_{-\mu}|^2)\tilde{a}_{\mu}$$
$$+ (\frac{1}{2}\Delta f_{\mu} + f_0 - f_p)a_{\mu} - i\frac{\kappa_{\mu}}{2}\tilde{a}_{\mu},$$
$$i\partial_t \tilde{a}_{-\mu} = -\gamma a_0^2 \tilde{a}_{\mu}^* - \gamma(|\tilde{a}_{-\mu}|^2 + 2|a_0|^2 + 2|\tilde{a}_{\mu}|^2)\tilde{a}_{-\mu}$$
$$+ (\frac{1}{2}\Delta f_{\mu} + f_0 - f_p)\tilde{a}_{-\mu} - i\frac{\kappa_{-\mu}}{2}\tilde{a}_{-\mu}. \tag{6}$$

Equation (6) explicitly shows that the OPO operation depends on the pump detuning, $f_0 - f_p$, and the phase-mismatch parameter, $\Delta f_{\mu}$.

Values of the relative linewidth parameters for the relevant signal-idler pairs are specified in Supplementary Table I. Uncertainties in the parameter selection for numerical modelling come from the absence of experimental data on the loss values in the proximity of the signal and idler fields and temperature-induced fluctuations of the refractive index impacting phase-matching conditions.

Pulse envelopes, $A_p$, $A_s$ and $A_i$, corresponding to the well-separated pump, signal and idler spectra can be reconstructed from the modal amplitudes as

$$A_p = \sum_{\mu=-N}^{N} a_{\mu}e^{i\mu\theta},$$
$$A_s = e^{-i\zeta_{\mu_0}t}\tilde{A}_s, \quad \tilde{A}_s = \sum_{\mu=\mu_0-N}^{\mu_0+N} \tilde{a}_{\mu}e^{i\mu\theta},$$
$$A_i = e^{i\zeta_{\mu_0}t}\tilde{A}_i, \quad \tilde{A}_i = \sum_{\mu=-\mu_0-N}^{-\mu_0+N} \tilde{a}_{\mu}e^{i\mu\theta}, \tag{7}$$

where $\mu_0$ and $N$ can be chosen, e.g. as $\mu_0 = 253$ and $N = 100$. We have now replaced the angular variable $\vartheta$ with $\theta = \vartheta - D_{1s}t$ rotating with repetition rate at the signal frequency and will approximate dispersions only upto the second-order terms. Equations for three envelopes, although their use for modelling comb dynamics is left for future work, allow for an understanding of the hyperparametric soliton generation at a qualitative level and contrast it with the previously reported parametric solitons[31–33]:

$$i\partial_t A_p = -i(D_{1p} - D_{1s})\partial_{\theta}A_p - \frac{1}{2}D_{2p}\partial_{\theta}^2 A_p - 2\gamma\tilde{A}_s\tilde{A}_i A_p^*$$
$$+ (\Delta_p - \gamma|A_p|^2)A_p - i\frac{\kappa_0}{2}(A_p - \sqrt{b\mathcal{W}}), \tag{8}$$

$$i\partial_t \tilde{A}_s = -\frac{1}{2}D_{2s}\partial_{\theta}^2\tilde{A}_s - \gamma A_p^2\tilde{A}_i^*$$
$$+ (\Delta_s - \gamma|\tilde{A}_s|^2)\tilde{A}_s - i\frac{\kappa_{\mu_0}}{2}\tilde{A}_s, \tag{9}$$

$$i\partial_t \tilde{A}_i = -i(D_{1i} - D_{1s})\partial_{\theta}\tilde{A}_i - \frac{1}{2}D_{2i}\partial_{\theta}^2\tilde{A}_i - \gamma A_p^2\tilde{A}_s^*$$
$$+ (\Delta_i - \gamma|\tilde{A}_i|^2)\tilde{A}_i - i\frac{\kappa_{-\mu_0}}{2}\tilde{A}_i, \tag{10}$$

where we omitted higher-order dispersion, and introduced short-hand notations for effective detunings incorporating pump detuning, phase-mismatch and nonlinear cross-phase modulation terms,

$$\Delta_p = f_0 - f_p - 2\gamma(|\widetilde{A}_s|^2 + |\widetilde{A}_i|^2), \tag{11}$$

$$\Delta_s = \frac{1}{2}\Delta f_{\mu_0} + f_0 - f_p - 2\gamma(|A_p|^2 + |\widetilde{A}_i|^2), \tag{12}$$

$$\Delta_i = \frac{1}{2}\Delta f_{\mu_0} + f_0 - f_p - 2\gamma(|A_p|^2 + |\widetilde{A}_s|^2). \tag{13}$$

Since, in the hyperparametric soliton regime, the pump and idler fields are quasi-CW, the $\Delta_s$ parameter can be approximately treated as a quasi-constant, $\partial_\theta \Delta_s \approx 0$, and equations for $A_p(t, \theta)$ and $\widetilde{A}_i(t, \theta)$ can be approximated by equations for $a_0(t)$ and $\widetilde{a}_{-\mu_0}(t)$. Then, Eq. (9) takes the form of the generalised Lugiato-Lefever model, where the four-wave mixing term, $A_p^2 \widetilde{A}_i^* \approx a_0^2 \widetilde{a}_{-\mu_0}^*, \partial_\theta(A_p^2 \widetilde{A}_i^*) \approx 0$, plays the role of an effective pump, implicitly parametrised by the actual pump detuning, $f_0 - f_p$, and power, $\mathcal{W}$. It is important to note that, even in this approximation, the effective pump in the signal equation, Eq. (9), is not an independent parameter like the pump term in the LLE model[2]. Instead, it is a variable whose stationary values and evolution are determined by the pump, $a_0(t)$, and idler amplitudes, $\widetilde{a}_{-\mu_0}(t)$, which satisfy additional equations. The normalised repetition rate differences and dispersion values are $(D_{1p} - D_{1s})/\frac{1}{2}\kappa_0 = -2.2126$, $(D_{1i} - D_{1s})/\frac{1}{2}\kappa_0 = 2.8199$, $D_{2p}/\frac{1}{2}\kappa_0 = 0.0052$, $D_{2s}/\frac{1}{2}\kappa_0 = 0.0085$ and $D_{2i}/\frac{1}{2}\kappa_0 = -0.0699$.

Eqs. (8)–(10) are transformed to the equations for the degenerate Kerr OPO used in Ref. 33 to demonstrate parametric solitons, when $A_p$ is replaced with $A_s$, while $\widetilde{A}_s$ and $\widetilde{A}_i$ become two pump fields $A_{p1}$ and $A_{p2}$, so that the external pump term, $\sqrt{b\mathcal{W}}$, is moved from Eq. (8) to Eqs. (9), (10). $A_{p1}$ and $A_{p2}$ can be approximated as quasi-CW fields[33], $\partial_t(A_{p1}A_{p2}) \approx 0$, so that Eq. (8) becomes a parametric Ginzburg-Landau equation,

$$i\partial_t A_s = -\frac{1}{2}D_{2s}\partial_\theta^2 A_s - 2\gamma A_{p1}A_{p2}A_s^* + (\Delta_s - \gamma|A_s|^2 - i\frac{\kappa_0}{2})A_s, \tag{14}$$

possessing explicit bright soliton solutions on zero background[32–35,38–40].

## Data availability
The raw data comprise high-volume measurements that require specialist processing and are not deposited in a public repository but can be made available from the corresponding authors upon request.

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

## Acknowledgements

This work was supported by a joint project between Research Ireland (SFI Grant No. 23/EPSRC/3920) and the UK Engineering and Physical Sciences Research Council (EPSRC Grant No. EP/X040844/1). Other support was received from CONNECT Centre (Grant No. 13/RC/2077.P2), the Royal Society (Grant No. IES/R3/223225), and Leading Innovation and Entrepreneurship Teams of Zhejiang (Grant No. 2023R01011).

## Author contributions

H.W. performed the device design and measurements with the assistance from M.A., E.H.K., L.W., V.K., Q.W. and W.G. X.J. and T.J.K. fabricated the microresonators. D.V.S. developed theory and performed numerical simulations of OPO operation and solitons. H.W., J.F.D. and D.V.S. wrote the manuscript text. D.V.S. and J.F.D. conceptualised and supervised the project. All authors commented on the manuscript.

## Competing interests

The authors declare no competing interests.
