## [Transparent Peer Review file · Nature Communications]

Hyperparametric solitons in nondegenerate optical parametric oscillators

Corresponding Author: Professor Dmitry Skryabin

Version 0:

Reviewer comments:

Reviewer #1

(Remarks to the Author)

The manuscript entitled “Hyperparametric solitons in nondegenerate optical parametric oscillators” by Haizhong Weng et al. reports the demonstration of solitons in a silicon-nitride microresonator-based nondegenerate OPO system, which by pumping a relatively low-Q resonance at 191 THz, they were able to excite a signal soliton comb centered around 240 THz as well as an idler comb at 140THz. This work is of interest to the microcomb and optical frequency comb community as it provides a way of generating optical frequency combs at wavelengths away from the pump wavelength, where the signal/idler comb generation wavelength can be engineered by choosing phase matching parameters. While the study is very interesting and carried out carefully, I strongly recommend that the authors to include the experimental demonstration of the control of the signal/idler comb generation wavelength by adjusting the phase matching parameters as indicated in the simulation results in the supplementary information Fig. S1. This would be very helpful as it would highlight the control over of the signal/idler comb generation wavelength. That being said, I believe that the manuscript is highly relevant to the Nature Communications audience.

Some minor comments are:

- 1) Can the authors include information about the limitations on the wavelength range for which a signal/idler optical frequency comb could be generated with such a silicon-nitride microresonator-based nondegenerate OPO system?
- 2) Can the authors include information on the trade-offs if one would want to generate broader signal/idler optical frequency combs?

Reviewer #2

(Remarks to the Author)

This manuscript reports the first experimental observation and theoretical description of hyperparametric solitons in a nondegenerate Kerr microresonator optical parametric oscillator (OPO). By pumping a Si₃N₄ microresonator in the C-band, the authors generate a three-color frequency comb, with a soliton state present in the far-detuned O-band signal. The key novelty is that these solitons at the signal frequency are generated from a nondegenerate Kerr OPO process, distinct from previously reported degenerate Kerr parametric solitons that require two pumps.

While the main claims are well-supported, several aspects require clarification, additional quantitative data, and expanded discussion before the paper is ready for publication.

One notable feature in the results is that the signal power is consistently higher than the idler power. The authors should explain the physical origin of this asymmetry. Furthermore, could the device be designed such that the idler wave hosts the soliton instead of the signal? Such a demonstration would broaden the scope and applicability of the work.

A key advantage of nondegenerate OPOs is their potential for wide spectral tunability. However, the present results show soliton generation only at a fixed signal/idler wavelength pair determined by the resonator geometry and pump mode. While Supplementary Fig. S2 demonstrates CW OPO tuning, soliton comb tunability is not demonstrated. If such data exist, they should be included; if not, the claims about demonstrated tunability should be tempered and framed as a potential capability

for future exploration. Additionally, the statement that solitons were “reproducibly observed in different resonators” would be more impactful if quantified, for example by reporting a yield such as “observed in X out of Y tested devices with similar geometry.”

The manuscript should clarify whether the reported pulse duration was estimated from the optical spectrum—assuming a transform-limited pulse—or directly obtained from numerical simulations.

For practical applications, it would also be useful to report absolute optical powers in each band (pump, signal, idler) and the total conversion efficiency from pump to signal/idler, in both the CW OPO and soliton regimes.

In Fig. 5(b), cases (i)–(iv) show no RF beat notes, unlike case (v). The authors should explain why no RF features are visible—whether this reflects complete coherence, or is instead due to measurement limitations.

Does the mismatch in group velocity between the pump, signal, and idler beams affect soliton formation?

Recent work on OPO soliton crystals (arXiv:2504.04788, 2025) reports multiple possible phase states of OPO solitons in crystals, which significantly influence comb structures.

The authors should discuss whether similar behavior was observed in their simulations and experiments. Has the author observed the biphasic soliton or crystals in this OPO configuration?

A pump–detuning phase diagram, distinguishing different operational states such as CW OPO, hyperparametric soliton, multi-soliton, and modulation instability regimes, would greatly aid in understanding the stability landscape of the system.

Reviewer #3

(Remarks to the Author)

Please see attached my report. Thank you

Version 1:

Reviewer comments:

Reviewer #1

(Remarks to the Author)

The authors have address my comments. I support the publication of the manuscript in Nature Communications.

Reviewer #2

(Remarks to the Author)

The manuscript has substantially improved. The rebuttal adds coherence and noise measurements, repetition-rate measurements, corrects the dispersion/FSR extraction, and reports absolute power and conversion efficiency.

Two focused items remain:

(1) Group-velocity mismatch. The response is still vague. Please quantify GVM and report (β_{1g}) or (n_g) at the pump/signal/idler centers.

(2) Pulse-width metric and experimental cross-check. State unambiguously whether you use intensity FWHM; if $1/e$ or $1/e^2$ widths appear, convert them to FWHM.

Reviewer #3

(Remarks to the Author)

The authors have significantly improved their manuscript and I commend them for providing new experimental data. The repetition rate measurements, the heterodyne beat measurements, and the additional data showing that the soliton forms at the signal frequency rather than being a pump-centered DKS spectrally translated via XPM and OPO. The noise measurements are also now convincing.

However, as stated in my first report, the authors do not demonstrate a new soliton regime but rather a new method for generating the driving field. The signal equation takes the form of a generalized Lugiato-Lefever equation in which the four-wave mixing term acts as an effective pump. Since this driving term is dominated by CW components from the nondegenerate OPO, the resulting signal soliton behaves as a standard DKS. Furthermore, coupled-LLE systems describing multi-color soliton dynamics have been studied extensively in the literature, as cited in my first report. This is not the first theoretical framework of its kind, even if the authors dismissed most prior work by claiming a different regime despite employing the same equation. The novelty lies in the pumping mechanism, which enables DKS generation outside the spectral window of the main pump, not in the soliton physics itself.

Does this distinction diminish the importance of their findings? I think not. Their work will be of interest for the community. Should this disagreement prevent publication? Again, I think not. This represents an opinion disagreement that time and the

community will resolve.

A minor point on "quasi-CW": the experimental data show comb teeth around the pump and idler, indicating that these fields are not physically CW. Otherwise, no comb would form around them, and only a single tone would appear in the experimental spectra. These waves are indeed dominated by the CW tone, which justifies the approximation of the parametric driving field of the DKS. However, these XPM-induced additional comb teeth exist and cannot be dismissed from experimental observation.

Given the solid results and overall contribution, I recommend publication in Nature Communications with the reservation that the authors address my comments either as a response or modifications.

We thank all reviewers for their careful consideration and for their comments on our manuscript. Below, we provide point-by-point replies to all comments, explaining the revisions made and the new work performed to address them.

In particular, we have performed and presented new experiments on soliton tunability, soliton noise measurements, and repetition rate measurements, and added new spectral data and linewidth and dispersion measurements. All new additions to the main manuscript text and supplementary document are highlighted in red font. The manuscript text and supplementary document lines are numbered, which further facilitates referencing in the reply letter.

Reviewer #1 (Remarks to the Author):

The manuscript entitled “Hyperparametric solitons in nondegenerate optical parametric oscillators” by Haizhong Weng et al. reports the demonstration of solitons in a silicon-nitride microresonator-based nondegenerate OPO system, which by pumping a relatively low-Q resonance at 191 THz, they were able to excite a signal soliton comb centered around 240 THz as well as an idler comb at 140THz. This work is of interest to the microcomb and optical frequency comb community as it provides a way of generating optical frequency combs at wavelengths away from the pump wavelength, where the signal/idler comb generation wavelength can be engineered by choosing phase matching parameters. While the study is very interesting and carried out carefully, I strongly recommend that the authors to include the experimental demonstration of the control of the signal/idler comb generation wavelength by adjusting the phase matching parameters as indicated in the simulation results in the supplementary information Fig. S1. This would be very helpful as it would highlight the control over of the signal/idler comb generation wavelength. That being said, I believe that the manuscript is highly relevant to the Nature Communications audience.

Response:

We thank the reviewer for the positive evaluation of our work and for highlighting its novelty and rigor. We agree that providing more experimental evidence of the tunability of the signal/idler comb generation wavelengths via phase-matching control is highly relevant and would significantly strengthen the manuscript. To address this point, we have conducted additional measurements and included the corresponding data in the revised manuscript.

In particular, we have updated Supplementary Fig. S4 to include new experimental results that directly address this point. In Fig. S4(b), we show hyperparametric single-solitons generated in the same resonator (i.e., the one used in the main text), where we tuned the pump wavelength to excite a sequence of different TM₀₀ modes. These experiments confirmed, respectively, the tunability of the signal and idler frequencies by varying the phase-matching wavelengths. Furthermore, Fig. S4(a) shows results from three other resonators with different ring widths and the same thickness, pumped at approximately 191.1 THz, demonstrating that dispersion engineering enables frequency tuning of the hyperparametric solitons when the pump frequency remains

constant.

New paragraphs in the main text (line 322) and in the supplementary material document (sub-section IV) describe these new results in detail for readers.

Minor comments:

1) Can the authors include information about the limitations on the wavelength range for which a signal/idler optical frequency comb could be generated with such a silicon-nitride microresonator-based nondegenerate OPO system?

Response:

To address this we have added a new paragraph in the main text, see line 475,

“Experimentally, we have observed the OPO comb spectra with signal and idler tuned between 230 and 249 THz (1.3-1.2 μm) and 138-150 THz (2.2-2 μm), respectively. At the extremities of this range, a broadband OPO comb was generated but not a soliton state due to the unoptimized coupling rate. While dispersion engineering allows extremely broad-band phase matching [22], the existence of multi-colour solitons imposes its own constraints on the balance of dispersion and quality factors for which a complete understanding should be the subject of future work.”

Overall, our revised manuscript provides new and solid evidence of the soliton’s signal/idler components’ tunability via dispersion engineering or pump frequency tuning. At the same time, we refrain from making unjustified claims which could overestimate the tunability range of hyperparametric solitons in the devices we are dealing with in this work. In future work, a reverse-design approach can help identify an optimal range of Q factors controlled by the waveguide-resonator gap and by simultaneous broadband dispersion engineering.

2) Can the authors include information on the trade-offs if one would want to generate broader signal/idler optical frequency combs?

Response:

This question echoes the previous one and the above response, but shifts emphasis to the comb bandwidth. A broader signal-idler frequency comb can be achieved by increasing the pump power.

This is now addressed by new experimental data in Fig. S5 (supplementary document, section V, line 174) and by the new paragraph starting in line 446 of the main text:

“We also observed the generation of spectrally broad signal and idler combs at relatively high pump powers, see Fig. S5 in the Supplementary Information. For large

detuning, a sparse Turing-pattern-like OPO comb has a line separation of 11 FSR, which then gradually transitions to combs with broader, densely filled spectra. The pump-only Turing-pattern-like comb with a 28 FSR line-to-line separation is seen at the end of this experimental sequence. This level of pump power already takes us outside the hyperparametric soliton range, while pure Kerr solitons can be found numerically at these powers for red detuning. They are, however, not observed experimentally due to the lack of measures to control thermal effects in our setup.”

Reviewer #2 (Remarks to the Author):

This manuscript reports the first experimental observation and theoretical description of hyperparametric solitons in a nondegenerate Kerr microresonator optical parametric oscillator (OPO). By pumping a Si₃N₄ microresonator in the C-band, the authors generate a three-color frequency comb, with a soliton state present in the far-detuned O-band signal. The key novelty is that these solitons at the signal frequency are generated from a nondegenerate Kerr OPO process, distinct from previously reported degenerate Kerr parametric solitons that require two pumps. While the main claims are well-supported, several aspects require clarification, additional quantitative data, and expanded discussion before the paper is ready for publication.

1. One notable feature in the results is that the signal power is consistently higher than the idler power. The authors should explain the physical origin of this asymmetry. Furthermore, could the device be designed such that the idler wave hosts the soliton instead of the signal? Such a demonstration would broaden the scope and applicability of the work.

Response:

The asymmetry mainly arises from coupling-loss dispersion, leading to different total Q factors for the signal and idler. It is critical that neither signal nor idler Q factors are too low. In our case, the lowest Q is at the pump frequency. Since the signal is gaining energy from a product of the pump and idler squared, a sufficiently high Q for the idler and hence the idler's power in the central component (not in the idler's comb) is critical for the generation of high bandwidth signal combs and also the high-power monochromatic central signal component. One should bear in mind that in the original manuscript, we measured linewidth only for the pump comb; now we have added measurements for the signal, but a suitable laser to measure the idler linewidth is not available to us. This leaves some uncertainty about the actual value of the idler's Q. In the Supplementary Fig. S2b, the idler intensity shows an increasing trend with the increase of frequency, which is consistent with the increasing trend of the Q factors shown in Fig. 2d. In the present 200-GHz SiN system, we notice that the idler always has a normal dispersion (see Fig. S1b), which hinders the powers in its sidebands, but not of its central component. Future work will explore new designs with variations in the ring radius and cross section, or new platforms (such as AlN or Ta₂O₅), to determine whether a long-wavelength idler wave could exhibit anomalous dispersion and host soliton generation.

To comment on this, we have added new text in lines 259-269 of the main manuscript.

2. A key advantage of nondegenerate OPOs is their potential for wide spectral tunability. However, the present results show soliton generation only at a fixed signal/idler wavelength pair determined by the resonator geometry and pump mode. While Supplementary Fig. S2 demonstrates CW OPO tuning, soliton comb tunability is not demonstrated. If such data exist, they should be included; if not, the claims about demonstrated tunability should be tempered and framed as a potential capability for future exploration. Additionally, the statement that solitons were “reproducibly observed in different resonators” would be more impactful if quantified, for example by reporting a yield such as “observed in X out of Y tested devices with similar geometry.”

Response:

We also thank this reviewer for the positive evaluation of our work and for highlighting its novelty and rigor. The tunability issue was also brought up by the 1st referee and to a significant degree addressed by us above. In particular, the revised Supplementary Fig. S4 demonstrates new experimental results on the tunability of the signal frequency by tuning the pump frequency and changing the resonator geometry, see for details and text edits our reply to referee 1 (line 322 of the main text and section IV of the supplementary).

3. The manuscript should clarify whether the reported pulse duration was estimated from the optical spectrum—assuming a transform-limited pulse—or directly obtained from numerical simulations.

Response:

The captions of Figs. 4 and 5 state that they are simulated pulse profiles. The numerical spectra match the experimental ones, and therefore, one can indeed rely on the numerical predictions of pulse durations.

4. For practical applications, it would also be useful to report absolute optical powers in each band (pump, signal, idler) and the total conversion efficiency from pump to signal/idler, in both the CW OPO and soliton regimes.

Response:

Lines 259 and 284 now state: *“The output powers of the pump and signal in Fig. 3b are 17.4 dBm and 5.4dBm, respectively, corresponding to an on-chip conversion efficiency of approximately 2%, which can be further improved by engineering the coupling rate and implementing temperature control.”*

and

“A pronounced feature of the soliton spectra is the dominance of the signal comb power over the pump and idler combs. Specifically, the total power of the pump comb accounts for only 4% of the signal comb power, excluding the central lines of both combs.”, respectively.

5. In Fig. 5(b), cases (i)–(iv) show no RF beat notes, unlike case (v). The authors should explain why no RF features are visible—whether this reflects complete coherence, or is instead due to measurement limitations.

Response:

The absence of RF beat notes in cases (i)–(iv) in Fig. 5(b) indeed indicates high coherence of the generated OPO soliton states. To avoid potential ambiguity, we employed an alternative method to directly characterise the coherence of these solitons by performing new heterodyne beat measurements between a comb line and an independent tunable laser (TSL). The new noise characterisation setup is shown in Fig. 3a, and its description is included in the Methods.

Line 470 now states: *“A single beat note near 1.5 GHz ($\textit{\$}_{beat}\$$) in heterodyne measurements shown in Fig. 5b confirms that the two-, three-, and four-soliton states are completely coherent. However, the five-, six-, and seven-soliton crystals are typically the breather states [53, 54].”*

6. Does the mismatch in group velocity between the pump, signal, and idler beams affect soliton formation?

Response:

Signal solitons and signal combs are primarily supported by the central components of the pump and idler fields, which suggests that the GV mismatch is not a key parameter. Though, of course, since all three fields have finite spectra, the GV mismatch will play its detrimental role if it is taken too far.

7. Recent work on OPO soliton crystals (arXiv:2504.04788, 2025) reports multiple possible phase states of OPO solitons in crystals, which significantly influence comb structures. The authors should discuss whether similar behavior was observed in their simulations and experiments. Has the author observed the biphasic soliton or crystals in this OPO configuration?

Response:

This mentioned arXiv manuscript work does not study the hyperparametric case (it studies degenerate OPO), thus we have not observed and reported these solitons. We

have added this citation, see Ref [61].

8. A pump–detuning phase diagram, distinguishing different operational states such as CW OPO, hyperparametric soliton, multi-soliton, and modulation instability regimes, would greatly aid in understanding the stability landscape of the system.

Response:

Such bifurcation diagrams are shown in Figs. 1a, 1b and 1d. In particular, Fig. 1d shows how CW-parametric states are excited at detunings below -2, then replaced by Turing non-solitonic combs for dimensionless detunings from -1.5 to -0.6, which then evolve into hyperparametric solitons, existing till just before zero detunings. Figs. 1a and 1b show detailed bifurcation diagrams for all existing (for given power) CW parametric states. This is explained in detail in lines 350-390, see line 368, in particular.

Reviewer #3 (Remarks to the Author):

Weng et al., in their manuscript "Hyperparametric solitons in nondegenerate optical parametric oscillators," present a novel method to generate solitons by creating the driving pump field from a pump-degenerate OPO within the same microring resonator. Their theory is thorough, and I find this regime interesting and likely useful to the community. However, as detailed in my comments below, the authors overclaim their results. I do not believe this is a "new soliton regime" but rather a new way to generate the driving field that yields the same soliton dynamics as direct pumping. They also lack a thorough review of previous work that frames the state-of-the-art. Additionally, their experimental work, though of good quality, does not convincingly demonstrate the actual claimed hyperparametric soliton regime. A clear demonstration that the soliton originates from the idler, not the main pump, is missing. This is especially true given the differing comb regimes observed while detuning the main pump, which are different between experiment and theory and cannot be solely blamed on thermal bistability (in particular since the theory claims blue-detuning operation). Therefore, in its current form, I do not recommend this manuscript for publication in Nature Communications. I hope the authors find my comments useful to improve their manuscript and enhance their work, which, as stated, is of interest and could be suitable for publication in a revised form.

1. "Our current work fills this gap by presenting a new class of dissipative optical solitons, and highlighting critical differences with solitons in degenerate OPOs." and "Here, we report a class of solitons in microresonator [...]" are not exactly correct statements, and overall goes with the sentiment that the paper overclaims what they are demonstrated. Surely, the authors generate the driving field parametrically, which is new. Yet, as shown theoretically, the driving force in the signal equation $2 p a + i$ ($p = \text{pump}$, $i = \text{idler}$) is mainly driven by a CW term from the pump-degenerated OPO. Thus, the signal soliton, whether generated by a direct laser or a parametric CW process, results in the same soliton: cascading from the driving force with a strong pedestal from the out-of-phase pump part, clearly observed theoretically and experimentally. The authors also show in the method that the signal-idler degenerated

parametric $\chi(2)$ or $\chi(3)$ solitons in refs [31-33] differ significantly. In such case, the complex conjugate of the soliton wave acts in the driving force term, substantially modifying the soliton's parametrically driven dynamics and properties, creating the granted new "soliton regime." Unfortunately, this does not apply to the authors' system and should be reflected in their claim.

Response:

We have now supported our claims with a significant amount of new experimental data included in Supplemental Materials and with revisions of the main text. These changes are described in detail in our response to the two other reviewers and include demonstrating comb tunability via pump tuning and dispersion engineering (resonator geometry). Novelty of the solitons is clear, since these are 3-equation solitons. The pump field and idler field, which drive the signal, are dynamically varying detuning sensitive fields even in their simplest monochromatic approximations (even more so when the pump and idler host significant comb spectra), therefore, solitons are clearly of a novel type and are not described by the Lugiato-Lefever model. Connecting to the latter is useful, of course, to make the next step in our knowledge about these systems clearer to readers.

2. The authors omit a large body of work using parametric interactions to generate pulses (soliton, dark soliton, simultons, etc.). Surprisingly, multi-pumped parametric interaction with DKS is cited but not properly linked to the authors' work. As noted, the authors' system resembles a multi-pump system, with the main pump and CW OPO, where parametric interaction expands the frequency comb (ref [8-9]), generates bright-dark states by cross-phase modulation binding (ref 11), or nonlinear mixes with the soliton to create a parametric optical trapping (Moille et al PRL 134 (19), 193802, 2025). A recent complete theoretical description by Menyuk et al. 33 (10), 21824-21835 Optics Express 2025 models a multi-color system similarly to the authors' method but is uncited, making the claim "Since no present concept or theory explains our measurements," too strong, though I agree the system lacks thorough theoretical study. Linking this state-of-the-art with the authors' work would aid readers and clarify the pump and idler comb spectra, also described by XPM-induced effects in Stone et al., Phys. Rev. Applied 17, 024038, 2022. The authors should also cite single-pump work on comb generation and parametric interaction, as their claim "no microresonator nondegenerate OPO that generates soliton microcombs has been demonstrated to date" is at best inaccurate. They omit Yang et al., Communications Physics 3, 27 (2020) on satellite frequency combs (not soliton microcombs) and Anderson et al., Phys. Rev. X 13, 011040, 2023 (especially figure 7), where cavity geometry modulation enables other degree of freedom for phase matching enabling parametric interaction with dark solitons for coherent satellite comb generation. Though these systems differ from the authors', these omissions bias the manuscript and fail to represent the field's state-of-the-art.

Response:

We thank the reviewer for bringing to our attention a list of further papers on solitons in microresonators published in reputable journals. This particular PRL concerns synchronization, a large topic in its own right that falls outside our current scope. The

Optics Express paper by Menyuk, et al studies three equations with 2 pumps and, crucially, operates outside the hyperparametric regimes we uncovered, with most of the energy transferred into the free-running (i.e., no-pump) signal field. We have added a reference to this work (Ref [15]), as well as to the next one by Stone (Ref [45]) and to Yang's work (Ref. [46]). Anderson's work on dispersion modulation is very valuable, but it does not align closely enough with the scope of our work.

3. The experimental results do not convincingly show a hyperparametric soliton. Both the main pump and signal lie in the anomalous dispersion regime, which should allow either to generate a soliton, and it is rather unclear how the authors prove that the soliton is not a "main-pump soliton" which is then spectrally translated around the signal/idler through XPM.

From their simulations, the hyperparametric soliton should appear in the main-pump blue-detuned regime. In their simulation, although not really described, Figure 4-d shows that further detuning the main pump reveals additional regimes at the main pump where, I believe, a main-pump soliton may appear while the pump-degenerated OPO may persists.

Experimentally in Figure 3-c, they show the system passing through multiple nonlinear regimes, including modulation instability, before reaching a soliton step, which begs the question if the authors are still blue-detuned. More importantly, the simulation does not corroborate such MI state appearing before reaching the hyperparametric soliton. Since cross-phase modulation alone explains the creation and bonding of the secondary pumped wave (see ref 11 and Menyuk et al), how do the authors experimentally exclude generating a main-pump soliton simply shifted around the CW-OPO signal? Without detuning measurements or a clear theory comparison of the intermediate states, their experimental evidence does not prove the hyperparametric soliton.

Response:

We thank the reviewer for raising this point. Our manuscript presents a comprehensive set of supporting evidence that the signal component is the primary component of the hyperparametric soliton bearing the strongest frequency comb.

- a) Using experimental data, we have calculated the total power of pump comb (180-198 THz, excluding its central line) as 0.1 mW, which is only 4% of the signal comb power (225-258 THz, excluding the central signal line), see line 286: *"Specifically, the total power of the pump comb accounts for only 4% of the signal comb power, excluding the central lines of both combs."*
- b) At 425 mW of pump, we experimentally observed the OPO soliton, which transitions directly from the OPO primary comb (see the video at 00:18) without any intermediate states. The evolution ends by the transition from OPO solitons to a few lines near the signal/pump/idler. The difference between the experiment and the simulation is seen in Fig. 4d only for larger detunings (the right most data interval). The buildup of intracavity power becomes noticeable at these detunings, leading to strong thermal effects that are not compensated for in our experimental setup. If they were to be compensated, we indeed could see non-hyperparametric solitons, i.e., DKSSs, discussed in many

previous papers. In support of this, the lowest panel in Fig. S5 shows the onset of pure-pump Turing patterns (comb lines are separated by $28 \times \text{FSRs}$), which, with thermal compensation techniques on, would evolve to Kerr-like pump solitons, see new text in line 446, “We also observed the generation of spectrally broad..”

- c) To further support the claim of hyperparametric, i.e., non-DKS, soliton regime, we have also performed a new set of repetition rate measurements by injecting the OPO single-soliton microcomb into the cascaded intensity-phase modulators, which gave us 199.58 GHz repetition rate. This rate is much closer to the simulated FSR of the signal ($D_{1s}=199.75$ GHz) than that of the pump ($D_{1p}=198.99$ GHz). The results are presented in Figs. 3e to 3f and the new section added to the main text starting in Line 288: “Coherence of the soliton state was confirmed via...”

4. Related to the claim of experimental demonstration of hyperparametric soliton, figure 5 is not convincing. The multi-DKS simulation shows clear comb spectral asymmetry around the pump mode, absent experimentally except in the top case. Also, the soliton crystal simulation shows dark teeth that are not very dark, possibly from the hyperparametric soliton’s background bump. Yet, the authors show perfect soliton crystals, (where interestingly Cnoidal waves/Turing rolls, and soliton crystals can occur in the blue-detuned regime see Qi et al, *Optica* 6, 9, 2019). These discrepancies cast doubt that the soliton forms at the signal rather than at the pump and then is translated through XPM around the signal. The authors could compare direct (multi)soliton generation at the main pump with their observed spectra to help confirm a hyperparametric soliton.

Response:

Hyperparametric solitons are distinctly different from DKS, see our point 3(a) above and the Concept section in the main manuscript text. The spectral asymmetry around the pump, seen in Fig. 5, is small, but it is clearly observed in both modelling and experiment for single- and multisoliton states. For solitons composed of three combs all propagating with different group velocities, the asymmetries of the individual combs are to be expected. Our experiment and modelling show clear spectra corresponding to soliton crystals. The spectral periodicity of the dominant lines is perfect. Extremely weak secondary spectra can sometimes be present or absent in the modelling and occur due to minor offsets of soliton locations from a perfect $2\pi/N$ angles along the ring circumference. This is a very natural, noise- and higher-order dispersion-sensitive feature that does not cast any doubt on the simulation results. These weak spectral backgrounds are always present in the experimental data shown in Fig. 5. As the number of solitons in a crystal grows, the central component of the signal spectra drops towards the comb level (our spectra should be judged on the signal, not the pump or idler). This

effect is due to the background energy being taken by soliton cores. Stepping from the every-line soliton spectra to every 2nd, every 3rd and so on in the spectra is another clear signature of soliton crystals. Turing patterns usually pop up around their critical periodicity, which is often associated with much larger line-to-line spacings.

5. The noise measurements are not thorough enough and lack a lot of detail to be able to claim low-noise operation. The authors do not report any detected power. If no light or light below the NEP is sent to the photodiode, only the noise floor appears. Therefore, it would help to compare their measurement with a noisy state to show the (multi)-soliton regime low-noise operation. The authors also fail to explain the beat in the noise measurement, which is puzzling. The system is not described as a breather, and no spectral overlap among the three comb components appears (which is not stated but should not be on the same frequency grid, but rather offset from one another). Hence, without a better explanation of the noise measurement, the soliton regime demonstration remains unconvincing.

Response:

To address this point in the revised manuscript, we directly measured the soliton coherence by performing a heterodyne beat measurement between a selected comb line and an independent tunable laser, see Fig. 3d. The new noise characterization setup can be found in Fig. 3a.

Moreover, for single-soliton, other than the newly performed heterodyne measurement, we have also characterized the repetition rate measurement of the hyperparametric single-soliton by injecting it into the cascaded intensity-phase modulators, see new Figure S3 and new Section III in the supplementary document.

For multi-solitons and soliton crystals, we have added the noise measurement results discussion in line 470 of the main text and Fig. 5b.

6. The ring resonator's wide width allows for multi-mode operation at the signal. Because the sidewalls are not straight, avoided mode crossings could occur. This has been shown to enable high-power multi-mode family OPO generation (see ref 21) with minimal XPM-mediated loss channels (see Stone et al., Phys. Rev. Applied 17, 024038, 2022). The spectrum in Figure 3b is notably clean, resembling the normal-dispersion regime main pump (see 24), despite the main pump being in the anomalous regime and at very high power. The authors do not measure the mode in the O-Band, where lasers are directly available. Their claim that they work with the same mode family as the main pump relies on their dispersion model, which, given Figure 2e and the high measurement noise, is not convincing. I believe it is a crucial point to clarify as this could lead to significant discrepancy between the theory and experiment.

Response:

We thank the reviewer for raising this important point. We carefully reviewed the transmission data and found that some resonance wavelengths were extracted incorrectly due to the strong interaction of the low-extinction ratio TM₀₀ modes with

the FP modes induced by the reflection between the waveguide facets. We have carefully checked the data and the corrected data are shown in Figs. 2d-2f. The FSR agreement between the simulation and measurement was greatly improved. Below is a zoom-in view of the one resonance case.

Also, to further confirm the measurements, we have acquired an O-band laser (1260-1360 nm) to measure the transmission. Corresponding data have been added to the Figs. 2d and 2e. The measurement and simulation also show a good overall agreement.

Additionally, we have pumped different modes and all of them can excite the OPO comb, as illustrated in Fig. S2b, which agrees with the simulation very well. Such OPO behavior are also demonstrated across several resonators with similar dimensions and all of them can support OPO operation with the same behavior which can be predicted by the simulation, as shown in Fig. S4. Modeling and measurements confirm that the signal and idler are the same mode family as the pump.

Minor remarks:

- The authors claim the pump is quasi-CW based on the azimuthal field profile. However, a comb clearly forms from XPM interaction with the hyperparametric soliton, altering the pump's azimuthal profile. It is mostly CW-driven but not quasi-CW; otherwise, no comb would appear.

Response:

Yes sure, but we see no difference between the mostly-CW and quasi-CW cases.

- It is unclear why the authors use a pulley configuration when the system's width is limited. A straight waveguide could offer similar coupling performance. Unless system losses must balance perfectly, since the parametric interaction occurs on-chip, the signal's Q_c affects only extraction efficiency, not parametric interaction efficiency.

Response:

Such pulley coupling was used to enhance the extraction efficiency and suppress the Raman scattering [7, 48] through the engineering of the Q factors. In the previous demonstrations, we found that only main-pump combs can be generated when the pump is in the under-coupling state. We intended to use the pulley coupling to enhance the coupling rate for the pump. While later, we noted that the system losses must be balanced carefully for soliton generation and pulley coupling offers more flexibility.

1. We thank Reviewer #1 for the positive evaluation of our revisions.

2. We thank Reviewer #2 for the positive evaluation of our revisions and address further comments below.

(1) Microresonators are most frequently characterised in terms of their repetition rates matching the Free Spectral Range at specific frequencies, which is a more natural and widely adopted description of the physics of microresonator frequency combs if compared to the group velocity language natural in the context of waveguides. Equation linking a group index, n_g , to the rep rate, D_1 , is well known, $D_1 = (c/n_g) / (2\pi R)$, where c is the vacuum light speed, c/n_g is the group velocity and R is the resonator radius. This, for example, gives 2.088 for the signal group index. Values of the repetition rates and their role in the observed effects are given and discussed in our manuscript; see, e.g., the text lines 247-249 and 676-678, and figure 2e. Furthermore, the experimental techniques we use measure the repetition rates directly, see section III in the supplemental materials. For these reasons, including values of the group indices and group velocities in the manuscript text would make the text appear disconnected from the standard methodology and techniques.

(2) By fitting the measured power spectra of the signal part of the soliton with a sech^2 function, the corresponding spectral width is found to be 2.0 THz. Assuming transform-limited pulses, this width corresponds to a pulse duration of 158 fs (FWHM). We have added the corresponding sentence in the main text (see lines 304-308): “By fitting the signal spectrum with a sech-squared function, we found that the soliton has a 3-dB bandwidth of 2 THz, giving a transform-limited pulse duration of 158 fs at full width half maximum.”

3. We thank Reviewer #3 for the positive evaluation of our revisions and address further comments below.

The reviewer further comments on their previous point that this is not a new soliton but “rather a new method for generating the driving field.” From the first version of our manuscript, we stated that our qualitative interpretation is in fact close to this point of view, see, in particular, see lines 667-675. However, this interpretation is not in contradiction and co-exists with the fact that the effective monochromatic pump field (entering the signal equation) is not a constant field as in the LLE model, but is a function of the true physical pump and idler fields satisfying two further equations. Therefore, even with this simplified interpretation, this is not an LLE system. We have added a new sentence to clarify this further in the Method section, lines 675-680: “It is important to note that, even in this approximation, the effective pump in the signal equation, Eq. (8b), is not an independent parameter like the pump term in the LLE model [4]. Instead, it is a variable whose stationary values and evolution are determined by the pump, $a_0(t)$, and idler amplitudes, $\tilde{a}_{-\mu_0}(t)$, which satisfy additional equations.” On the further point, we consistently describe the idler and pump components of the solitons as quasi-CW fields.

Response to the XPM comment from Referee 3:

Comment: "However, these XPM-induced additional comb teeth exist and cannot be dismissed from experimental observation."

Reply: This comment connects to the ones on the quasi-CW nature of the idler and pump fields. The XPM-induced effects the referee is referring to are reflected in our experimental and modelling data via additional comb teeth, and are explicitly mentioned next to Eqs. 11-13. The term "quasi-CW," as used extensively in our discussions, explicitly indicates that the central CW component of a given group of modes, e.g., clustered around the idler, is dominant within this group, where XPM and other effects induce the weaker comb teeth.

Weng et al., in their manuscript "Hyperparametric solitons in nondegenerate optical parametric oscillators," present a novel method to generate solitons by creating the driving pump field from a pump-degenerate OPO within the same microring resonator. Their theory is thorough, and I find this regime interesting and likely useful to the community. However, as detailed in my comments below, the authors overclaim their results. I do not believe this is a "new soliton regime" but rather a new way to generate the driving field that yields the same soliton dynamics as direct pumping. They also lack a thorough review of previous work that frames the state-of-the-art. Additionally, their experimental work, though of good quality, does not convincingly demonstrate the actual claimed hyperparametric soliton regime. A clear demonstration that the soliton originates from the idler, not the main pump, is missing. This is especially true given the differing comb regimes observed while detuning the main pump, which are different between experiment and theory and cannot be solely blamed on thermal bistability (in particular since the theory claims blue-detuning operation).

Therefore, in its current form, I do not recommend this manuscript for publication in Nature Communications. I hope the authors find my comments useful to improve their manuscript and enhance their work, which, as stated, is of interest and could be suitable for publication in a revised form.

- "Our current work fills this gap by presenting a new class of dissipative optical solitons, and highlighting critical differences with solitons in degenerate OPOs." and "Here, we report a class of solitons in microresonator [...]" are not exactly correct statements, and overall goes with the sentiment that the paper overclaims what they are demonstrated. Surely, the authors generate the driving field parametrically, which is new. Yet, as shown theoretically, the driving force in the signal equation $a_p^2 a_i^*$ (p = pump, i = idler) is mainly driven by a CW term from the pump-degenerated OPO. Thus, the signal soliton, whether generated by a direct laser or a parametric CW process, results in the same soliton: cascading from the driving force with a strong pedestal from the out-of-phase pump part, clearly observed theoretically and experimentally. The authors also show in the method that the signal-idler degenerated parametric $\chi^{(2)}$ or $\chi^{(3)}$ solitons in refs [31-33] differ significantly. In such case, the complex conjugate of the soliton wave acts in the driving force term, substantially modifying the soliton's parametrically driven dynamics and properties, creating the granted new "soliton regime." Unfortunately, this does not apply to the authors' system and should be reflected in their claim.
- The authors omit a large body of work using parametric interactions to generate pulses (soliton, dark soliton, similitons, etc.). Surprisingly, multi-pumped parametric interaction with DKS is cited but not properly linked to the authors' work. As noted, the authors' system resembles a multi-pump system, with the main pump and CW OPO, where parametric interaction expands the frequency comb (ref [8-9]), generates bright-dark states by cross-phase modulation binding (ref 11), or nonlinear mixes with the soliton to create a parametric optical trapping (Moille et al PRL 134 (19), 193802, 2025). A recent complete theoretical description by Menyuk et al. 33 (10), 21824-21835 Optics Express 2025 models a multi-color system similarly to the authors' method but is uncited, making the claim "Since no present concept or theory explains our measurements," too strong, though I agree the system lacks thorough theoretical study. Linking this state-of-the-art with the authors' work would aid readers and clarify the pump and idler comb spectra, also described by XPM-induced effects in Stone et al., Phys. Rev. Applied 17, 024038, 2022. The authors should also cite single-pump work on comb generation and parametric interaction, as their claim "no microresonator nondegenerate OPO that generates soliton microcombs has been demonstrated to date" is at best inaccurate. They omit Yang et al., Communications Physics 3, 27 (2020) on satellite frequency combs (not soliton microcombs) and Anderson et al., Phys. Rev. X 13, 011040, 2023 (especially figure 7), where cavity geometry modulation enables other degree of freedom for phase matching enabling parametric interaction with dark solitons for coherent satellite comb generation. Though these systems differ from the authors', these omissions bias the manuscript and fail to represent the field's state-of-the-art.
- The experimental results do not convincingly show a hyperparametric soliton. Both the main pump and signal lie in the anomalous dispersion regime, which should allow either to generate a soliton, and it is rather unclear how the authors prove that the soliton is not a "main-pump soliton" which is then spectrally translated around the signal/idler through XPM. From their simulations, the hyperparametric soliton should appear in the main-pump blue-detuned regime. In their simulation, although not really described, Figure 4-d shows that further detuning the main pump reveals additional regimes at the main pump where, I believe, a main-pump soliton may appear while the pump-degenerated OPO may persist. Experimentally in Figure 3-c, they show the system passing through multiple nonlinear regimes, including modulation instability, before reaching a soliton step, which begs the question if the authors are still blue-detuned. More importantly, the simulation does not corroborate such MI state appearing before reaching the hyperparametric soliton. Since cross-phase modulation alone explains the creation and bonding of the secondary pumped wave (see ref 11 and Menyuk et al), how do the authors experimentally exclude generating a main-pump soliton simply shifted around the CW-OPO signal? Without detuning measurements or a clear theory comparison of the intermediate states, their experimental evidence does not prove the hyperparametric soliton.

- Related to the claim of experimental demonstration of hyperparametric soliton, figure 5 is not convincing. The multi-DKS simulation shows clear comb spectral asymmetry around the pump mode, absent experimentally except in the top case. Also, the soliton crystal simulation shows dark teeth that are not very dark, possibly from the hyperparametric soliton's background bump. Yet, the authors show perfect soliton crystals, (where interestingly Cnoidal waves/Turing rolls, and soliton crystals can occur in the blue-detuned regime see Qi et al, Optica 6, 9, 2019). These discrepancies cast doubt that the soliton forms at the signal rather than at the pump and then is translated through XPM around the signal. The authors could compare direct (multi)soliton generation at the main pump with their observed spectra to help confirm a hyperparametric soliton.
- The noise measurements are not thorough enough and lack a lot of detail to be able to claim low-noise operation. The authors do not report any detected power. If no light or light below the NEP is sent to the photodiode, only the noise floor appears. Therefore, it would help to compare their measurement with a noisy state to show the (multi)-soliton regime low-noise operation. The authors also fail to explain the beat in the noise measurement, which is puzzling. The system is not described as a breather, and no spectral overlap among the three comb components appears (which is not stated but should not be on the same frequency grid, but rather offset from one another). Hence, without a better explanation of the noise measurement, the soliton regime demonstration remains unconvincing.
- The ring resonator's wide width allows for multi-mode operation at the signal. Because the sidewalls are not straight, avoided mode crossings could occur. This has been shown to enable high-power multi-mode family OPO generation (see ref 21) with minimal XPM-mediated loss channels (see Stone et al., Phys. Rev. Applied 17, 024038, 2022). The spectrum in Figure 3b is notably clean, resembling the normal-dispersion regime main pump (see 24), despite the main pump being in the anomalous regime and at very high power. The authors do not measure the mode in the O-Band, where lasers are directly available. Their claim that they work with the same mode family as the main pump relies on their dispersion model, which, given Figure 2e and the high measurement noise, is not convincing. I believe it is a crucial point to clarify as this could lead to significant discrepancy between the theory and experiment.

Minor remarks:

- The authors claim the pump is quasi-CW based on the azimuthal field profile. However, a comb clearly forms from XPM interaction with the hyperparametric soliton, altering the pump's azimuthal profile. It is mostly CW-driven but not quasi-CW; otherwise, no comb would appear.
- It is unclear why the authors use a pulley configuration when the system's width is limited. A straight waveguide could offer similar coupling performance. Unless system losses must balance perfectly, since the parametric interaction occurs on-chip, the signal's Q_c affects only extraction efficiency, not parametric interaction efficiency.